# CD8^+^ T Cell Phenotype and Function in Childhood and Adult-Onset Connective Tissue Disease

**DOI:** 10.3390/ijms231911431

**Published:** 2022-09-28

**Authors:** Anna Radziszewska, Zachary Moulder, Elizabeth C. Jury, Coziana Ciurtin

**Affiliations:** 1Centre for Adolescent Rheumatology Versus Arthritis at University College London (UCL), University College London Hospital (UCLH), Great Ormond Street Hospital (GOSH), London WC1E 6JF, UK; 2Centre for Rheumatology Research, Division of Medicine, University College London, London WC1E 6JF, UK; 3University College London Medical School, University College London, London WC1E 6DE, UK

**Keywords:** CD8^+^ T cells, SLE, Sjögren’s syndrome, systemic sclerosis, scleroderma, polymyositis, dermatomyositis, connective tissue disease

## Abstract

CD8^+^ T cells are cytotoxic lymphocytes that destroy pathogen infected and malignant cells through release of cytolytic molecules and proinflammatory cytokines. Although the role of CD8^+^ T cells in connective tissue diseases (CTDs) has not been explored as thoroughly as that of other immune cells, research focusing on this key component of the immune system has recently gained momentum. Aberrations in cytotoxic cell function may have implications in triggering autoimmunity and may promote tissue damage leading to exacerbation of disease. In this comprehensive review of current literature, we examine the role of CD8^+^ T cells in systemic lupus erythematosus, Sjögren’s syndrome, systemic sclerosis, polymyositis, and dermatomyositis with specific focus on comparing what is known about CD8^+^ T cell peripheral blood phenotypes, CD8^+^ T cell function, and CD8^+^ T cell organ-specific profiles in adult and juvenile forms of these disorders. Although, the precise role of CD8^+^ T cells in the initiation of autoimmunity and disease progression remains to be elucidated, increasing evidence indicates that CD8^+^ T cells are emerging as an attractive target for therapy in CTDs.

## 1. Introduction

Systemic autoimmune conditions are multifactorial and result from inappropriate recognition of antigens and dysregulation of immunoregulatory processes. They are characterized by aberrant immune cell activation and failure to eliminate self-reactive lymphocytes leading to tissue damage [1]. Although the role of CD8^+^ T cells in autoimmune conditions has been relatively neglected in comparison to other immune cell types, increasing evidence indicates that CD8^+^ T cells contribute to progression and regulation of autoimmunity [2]. Here, we summarize the current evidence examining the role of CD8^+^ T cells in systemic lupus erythematous (SLE), systemic sclerosis (SSc), Sjögren’s syndrome (SS), polymyositis (PM) and dermatomyositis (DM) with special focus on commonalties in adult and juvenile connective tissue diseases (CTDs) in CD8^+^ T cell peripheral blood phenotypes, CD8^+^ T cell function, and organ-specific CD8^+^ T cell profiles.

### 1.1. CD8^+^ Cytotoxic T Cells

CD8^+^ T cells are an important component of the adaptive immune system involved in infection control and cancer immunosurveillance. They express the T cell receptor (TCR) and the CD8 co-receptor to recognize foreign peptides that are presented on major histocompatibility complex class I (MHC-I) molecules on APCs (antigen presenting cells). Upon encountering antigen, naïve CD8^+^ cells differentiate into effector cells capable of entering tissues to destroy pathogen-infected or cancerous cells expressing the cognate peptide-MHC-I combination. Once antigen is cleared, the expanded effector population undergoes a contraction phase and the remaining cells differentiate into long-lived effector memory (EM) and central memory (CM) cells [3]. Defining features of these CD8^+^ T cell subpopulations and surface molecules used to identify them are shown in Figure 1 [4].

CD8^+^ T cells exert their effector function in three ways: by releasing granules filled with cytotoxic molecules, through Fas (CD95)/FasL (Fas ligand) interactions, and by releasing cytokines [5]. Typical cytotoxic CD8^+^ T cells (Tc1) express high levels of cytolytic enzymes (perforin, granzyme B) and proinflammatory and antiviral cytokines (interferon-γ (IFN-γ) and tumour necrosis factor-α (TNF-α)), which cooperate to destroy malignant or virally infected cells [6]. Besides Tc1, there are various functionally distinct subsets of effector CD8^+^ T cells distinguishable by their cytokine profile as outlined in Table 1 [6].

### 1.2. CD8^+^ Tregs

In addition to cytotoxic CD8^+^ T cells, a regulatory subset of CD8^+^ T cells (CD8^+^ Tregs) with immunosuppressive function has also been identified. In their regulatory role, CD8^+^ cells maintain immune homeostasis by preventing inappropriate T and B cell activation. Various combinations of markers have been used to characterize CD8^+^ Tregs in the blood, including forkhead box P3 (FoxP3^+^), CD122^+^, CD28^−^, CD45RC^low^, and human lymphocyte antigen E (HLA-E) restriction [16]. Although no reliable CD8^+^ Treg cell markers that can distinguish CD8^+^ Tregs from ordinary CD8^+^ T cells have yet been found, expression of the transcription factor FoxP3 remains the most sensitive marker of Tregs to date [17,18]. Heterogeneous populations with various phenotypes and properties have been described, however few naturally occurring human CD8^+^ Treg populations have thus far been shown to exist and the majority of Treg subsets have been induced ex vivo [16]. 

CD8^+^ Tregs have been shown to exert their immunosuppressive action in four ways [17]: Release of inhibitory cytokines such as IL-10 and transforming growth factor β (TGFβ) [19];Rendering APCs tolerogenic through downregulation of costimulatory molecules CD80 and CD86 on APCs and upregulation of immunoglobulin-like transcript (ILT)3 and ILT4 in APCs [20,21];Suppression of CD4^+^ T cells via cell contact dependent mechanisms involving TGFβ and cytotoxic T cell antigen 4 (CTLA-4) expression on the CD8^+^ Treg surface [19];Perforin mediated cytotoxicity against antigen activated follicular helper CD4^+^ T cells expressing Qa-1 (mouse equivalent of human HLA-E) [22].

Over the past several years there has been renewed interest in CD8^+^ Tregs. Nevertheless, the field remains controversial with conflicting evidence and some of the described subsets may have regulatory or cytotoxic properties depending on the context or indeed use direct cytotoxic killing as a mechanism of suppression. However, hope remains that with further characterization CD8^+^ Treg cells with immunosuppressive properties could be very useful in T cell-based therapies in CTDs.

### 1.3. CD8^+^ Trm Cells

More recently, specialized populations of CD8^+^ T cells which reside in non-lymphoid tissues have been identified. These cells have been named tissue resident T cells (Trm) and unlike conventional memory cells remain at sites of infection over long periods of time. CD8^+^ Trm cells have been found in numerous sites following infection including in the skin, gut, brain, salivary glands, stomach, kidney, and heart [23]. These cells do not traffic around the body; they are terminally differentiated with minimal turnover and do not survive well upon introduction to circulation [24,25]. CD8^+^ Trm cells in tissues can be identified by expression of the αE integrin (CD103) and CD69 [24]. This distinct subset of CD8^+^ T cells may play a role in promoting autoimmunity in susceptible individuals and help mediate organ damage in CTDs.

### 1.4. Systemic Lupus Erythematosus (SLE)

SLE is a complex, multisystem, autoimmune condition that is characterized by the production of autoantibodies to nuclear components. Although the aetiology of the disease is not completely understood, a combination of genetic predisposition and environmental factors are thought to contribute to disease pathogenesis. Disease presentation, clinical course, and outcome may vary considerably between individuals, age groups, and ethnicities. Any organ can be affected in SLE, with common involvement of the mucocutaneous, musculoskeletal, renal, and haematological systems. Other manifestations include neuropsychiatric, pulmonary, gastrointestinal, cardiovascular, and ocular involvement as well as pregnancy complications [26]. When SLE is diagnosed before the age of 18 (or earlier according to various categorisation systems), it is classified as juvenile systemic lupus erythematosus (JSLE). JSLE is characterized by more aggressive disease, widespread organ involvement, and worse outcomes compared to adult-onset disease [27,28]. While many aspects of immune dysfunction have been studied extensively in SLE, there is limited and at times contradictory evidence of how cytotoxic CD8^+^ T cells contribute to the disease. Both defective and enhanced cytotoxic CD8^+^ T cell function in SLE has been described [29,30]. Defective cytotoxic function in SLE may imply failure to remove autoreactive B cells which produce the antibody complexes that are pathognomonic of SLE [31], whereas enhanced CD8^+^ T cell cytotoxicity may result in generation of high amounts of nuclear autoantigen and propagation of disease through enhanced production of autoantibodies by activated B cells, as well as tissue damage of target organs by CD8^+^ T cells [30]. 

### 1.5. Sjögren’s Syndrome (SS)

SS is a CTD that primarily affects the salivary and lachrymal glands, but can affect any mucosal exocrine gland [32]. SS can be classified as primary SS (pSS) or secondary SS (sSS). sSS is diagnosed when it occurs in conjunction with other autoimmune conditions such as SLE, rheumatoid arthritis (RA) or SSc. Destruction of the exocrine gland epithelium by aberrant B and T cell responses is a feature of the disease leading to the hallmark symptoms of dryness. Common presentations of SS (xerostomia, keratoconjunctivitis sicca, skin and vaginal dryness, dry cough) relate to diminished exocrine gland function. Extra-glandular manifestations are the consequence of lymphocytic infiltrates and immune complex deposition in various organs such as skin, joints, lungs, liver, kidneys, blood vessels, while serological manifestations, often comprising of cytopenias, increased polyclonal IgG and low complement levels, reflect abnormalities related to immune activation and chronic inflammation [33]. When the disease starts during childhood, it is called juvenile Sjögren’s syndrome and it is characterised by different clinical presentation, with a predominance of systemic features, lymphadenopathy and parotitis, and less frequent reports of significant dryness [34,35]. Cytotoxic CD8^+^ T cells have been found in the glandular tissue in SS and may be responsible for destruction of lacrimal and salivary glands [36].

### 1.6. Systemic Scleroderma/Systemic Sclerosis (SSc)

SSc is characterized by a chronic and progressive course of pathogenic changes. Vascular damage, autoantibody generation, innate and adaptive immunity activation, and fibrosis of the skin and visceral organs are all pathogenic mechanisms that lead to irreversible scarring and eventual organ failure [37]. SSc has various patterns of presentation, and these are grouped into either: diffuse cutaneous systemic sclerosis (dcSSc) which affects both truncal and acral skin areas, or limited cutaneous systemic sclerosis (lcSSc), affecting predominantly the extremities (acral involvement). Rapid progression of fibrosis affecting the skin, lungs, kidneys, and other internal organs is more common in dcSSc while Raynaud’s phenomenon, pulmonary hypertension, with or without skin calcification, gastrointestinal disease, telangiectasias (CREST syndrome), are more commonly associated with lcSSc [37]. When the disease affects predominantly the skin without significant organ involvement, it is called localised scleroderma, which is a common presentation in children aged 16 or younger [38]. However, localised scleroderma is associated with poorer outcome than in adults, as extracutaneous involvement in children (seizures, visual impairment, hemiatrophy, arthropathy) are more prevalent, disease relapses more common, and disease duration longer than in adults [38]. Children can also suffer from SSc, but this tends to be less severe than in adults because children have less internal organ involvement, and better long-term outcome [39]. Effector CD8^+^ T cells from patients with SSc have been shown to exhibit strong cytotoxicity and produce excessive amounts of profibrotic interleukin-13 (IL-13) and subsequently lead to more severe forms of cutaneous fibrosis [40].

### 1.7. Polymyositis (PM) and Dermatomyositis (DM)

PM and DM are two subtypes of idiopathic inflammatory myopathies. Both PM and DM are characterized by a proximal muscular weakness and muscle inflammation. In DM this is also accompanied by erythematous skin changes [41]. Examples of these skin changes include: a heliotropic rash, Gottron’s papules, and subcutaneous calcification. Juvenile DM (JDM) is the most frequent idiopathic inflammatory myopathy in children with a similar presentation and pathology to the adult form [42]. A muscle biopsy is essential for distinguishing between different types of myopathies (inflammatory and metabolic). PM muscle pathology is characterized by endomysial inflammatory infiltration with CD8^+^ T cell invasion of healthy MHC-I expressing muscle fibres. DM muscle pathology features evidence of injury to capillaries and perifascicular microfibers while skin biopsy reveals CD4^+^ T cell infiltration [41]. Thus, immunopathology in PM appears to be driven by destruction of muscle fibres by cytotoxic CD8^+^ T cells, while in DM autoantibodies and B cells seem to play a more significant role [41].

## 2. Aims and Methodology

In this comprehensive review we aimed to evaluate the evidence for CD8^+^ T cell involvement in SLE, SS, SSc, PM and DM with particular emphasis on similarities between CD8^+^ T cell peripheral blood profiles, CD8^+^ T cell function, and CD8^+^ T cell organ profiles across different CTDs and across age. We used two search strategies to ensure that this review covers the most up to date literature related to adult studies and CD8^+^ T cell peripheral blood signatures without replicating the excellent reviews already published in this field [2,43,44,45], while also ensuring that we capture the literature related to juvenile disease phenotypes and tissue specific CD8^+^ T cell signatures because of the paucity of data in this field and lack of previous comprehensive reviews. Therefore, the CD8^+^ T cell peripheral blood profiles searches in adult diseases were limited to human studies published from 2000 to present, involving at least 10 participants per disease and control group. We included studies with well described methodology and detailed patient cohort characterisation which examined frequencies or absolute values of total CD8^+^ T cells or CD8^+^ T cell subpopulations and their correlation with clinical parameters. We focused on peripheral blood subsets that are detectable in humans ex vivo, excluding inducible CD8^+^ T cell subsets. For this reason, our review is limited to only one putative CD8^+^ Treg subset: CD8^+^CD28^−^ cells. Excellent reviews on CD8^+^ Treg populations in autoimmune diseases have been published in recent years, which cover the topic in great detail [17,46,47,48]. To capture CD8^+^ T cell studies in juvenile diseases and reports describing CD8^+^ T cell distribution in target organs, we extended the search to all published human studies, regardless of date of publication or number of participants. The literature search was conducted on PubMed using combinations of the following keywords: CD8^+^, cytotoxic T lymphocytes, SLE, Sjögren’s syndrome, scleroderma, systemic sclerosis, polymyositis, dermatomyositis, myositis.

## 3. Results

### 3.1. Altered CD8^+^ T cell Phenotype in Adult CTD

#### 3.1.1. SLE

Of all the reports identified in the literature since 2000, 43 papers described phenotypic differences in the total CD8^+^ T cell population or in CD8^+^ T cell subsets in adult-onset SLE (aSLE). Eight of these studies with the largest number of well-characterized participants are highlighted in Table 2. A further 19 studies chosen based on inclusion criteria outlined above, are presented in Appendix A. Six studies found no significant differences in total CD8^+^ T cell frequencies in peripheral blood [29,30,49,50,51,52], while several others [53,54,55,56,57,58,59] reported an increase in SLE compared to healthy controls (HC). Absolute CD8^+^ T counts and CD4/CD8 T cell ratios, on the other hand, appeared to be consistently reduced in SLE across studies [51,52,53,54,56,60] apart from one study [50] where no difference in absolute CD8^+^ T cell counts was observed. Neither the proportion of CD8^+^ T cells nor absolute CD8^+^ T counts seemed to correlate with clinical or serological disease activity in patients with aSLE. Several studies reported a reduction in CD8^+^ T cell memory [61], CM [29], or EM [29,49,61,62] T cell populations in SLE, in particular, in patients with high disease activity [29,49,62]. Terminally differentiated effector memory (TEMRA) T cell frequencies [30,63], on the other hand, have been shown by some to be increased in patients with active SLE and this increase has been associated with lupus nephritis (LN) [63]. An association between the increase in memory cells (both EM and CM) and poor prognosis in SLE has also been reported [64]. In contrast, a recent large study involving 143 Asian SLE patients and 49 HC found no statistically significant differences in proportions of CD8^+^ CM, EM or TEMRA populations in SLE [65]. 

The variability in results pertaining to CD8^+^ phenotyping in the blood in aSLE, could be attributed in part to heterogeneity of the disease. In support of this, recently another large study of 143 consecutive treatment-naïve aSLE patients in China has demonstrated that it was possible to stratify patients based on their lymphocyte subset proportions [66]. Using this approach, patients with high CD8^+^ T cell frequencies were found to exhibit a higher rate of LN.

In contrast to the conflicting CD8^+^ memory T cell phenotype data, there seems to be a consensus in the literature that there is an increase in activated human leukocyte antigen-DR (HLA-DR)^+^ [30,49,52,65,67] and CD38^+^ CD8^+^ T cells [56] in the peripheral blood in SLE, particularly in active disease. The increase in activated CD8^+^ T cells was not associated with treatment [52,53,67], but rather, appeared to be a feature of the disease.

**Table 2 ijms-23-11431-t002:** Altered peripheral blood CD8^+^ T cell phenotype in adults with SLE.

Author et al., Year [Ref]	Type of Study	N: Patients or Controls Age ^1^: Mean ± SD or Median (Range) or [IQR]	CD8^+^ T Cell Populations	Clinical Relevance
Lu Z. et al., 2020 [66]	Cross-sectional	N = 143 SLE 35 [26–48] N = 30 HC ASM	Increased % of CD8^+^ T cells in a subset of treatment naïve SLE patients identified by hierarchical cluster analysis.	The patient cluster with higher % of CD8^+^ T cells had higher incidence of LN (OR 2.85, CI 1.15–7.08, *p* = 0.024).
Lai Z-W. et al., 2018 [61]	Clinical trial	N = 40 SLE, Mean: 45.4 (18–71) N = 56 HC, Mean: 45.4 (18–71)	Increased % CD8^+^CD45RA^+^ naïve T cells in SLE vs. HC (*p* < 0.05) Decreased % CD8^+^CD45RO^+^ memory T cells in SLE vs. HC (*p* < 0.05) Decreased % CD8 EM (CD62L^−^CD197^−^) T cells in SLE vs. HC (*p* < 0.05).	Reduction in CD8^+^ memory T cells was reversed after 12 months of sirolimus treatment and was the strongest predictor of therapeutic response.
Kubo S. et al., 2017 [65]	Longitudinal prospective	N = 143 SLE 42.7 ± 16.4 N = 49 HC 45.6 ± 14.7	No differences in % of CD8^+^ T cell subpopulations (naïve, EM, CM, TEMRA). Increase in % of activated CD8^+^ T cells (CD3^+^CD8^+^CD38^+^HLA-DR^+^) in SLE vs. HC (*p* < 0.001).	Positive correlation between % activated CD8^+^ T cells and SLEDAI (r = 0.27, *p* < 0.01) and BILAG (r = 0.38, *p* < 0.01). Decrease in CD8^+^ activated T cells in active patients after 24-week treatment with cyclophosphamide, mycophenolate mofetil, or calcineurin, in addition to high-dose glucocorticoids (*p* < 0.01).
Comte D. et al., 2017 [29]	Cross-sectional	N = 45 SLE 41.3 (21–72) N = 41 HC ASM	No difference in % CD3^+^CD8^+^ T cells between SLE and HC. Decreased % CD8^+^ EM and CM in SLE vs. HC (*p* < 0.05). Increased % naïve CD8^+^ T cells in SLE vs. HC (*p* < 0.05). Reduction in perforin (*p* < 0.01) and granzyme B (*p* < 0.05) in SLE CD8^+^ T cells vs. HC. Reduced CD8^+^ T cell SLAMF7 expression in active SLE vs. HC (*p* < 0.001).	Decreased % CD8^+^ EM and CM T cells in active SLE vs. HC (*p* < 0.01 and *p* < 0.05, respectively). Increased % naïve CD8^+^ T cells in active SLE vs. HC (*p* < 0.001) and vs. inactive SLE (*p* < 0.05). Positive correlation between % naïve CD8^+^ T cells and SLEDAI score (r^2^ = 0.14, *p* = 0.01). Negative correlation between % TEMRA CCR7^−^CD45RA^+^ CD8^+^ T cells and SLEDAI score (r^2^ = 0.09, *p* = 0.05).
Zabinska M. et al., 2016 [68]	Cross-sectional	N = 54 SLE with LN SLEDAI ≤ 5: 32.7 ± 9.1 SLEDAI > 5: 37.9 ± 14.9 N = 19 HC 38.3 ± 14.1	Increased % and absolute counts of CD3^+^CD8^+^CD28^−^ T cells in SLE vs. HC (*p* < 0.001).	Positive correlation between SLEDAI and % CD3^+^CD8^+^CD28^−^ T cells (r = 0.281, *p* = 0.038). Increased % CD3^+^CD8^+^CD28^−^ T cells in SLE with active vs. inactive LN (*p* = 0.022).
Tulunay A. et al., 2008 [50]	Cross-sectional	N = 53 SLE 39 ± 12 N = 44 HC 37 ± 14	No difference in % or absolute CD8^+^ T cell counts in SLE vs. HC or in absolute CD8^+^CD28^+^ or CD8^+^CD28^−^ T cell counts. Decreased % CD28^−^ of CD8^+^ T cells in SLE vs. HC (*p* < 0.01). Increased % CD28^+^ of CD8^+^ T cells in SLE vs. HC (*p* < 0.01).	No association between absolute numbers of CD8^+^CD28^+^ or CD8^+^CD28^−^ T cells and SLEDAI or treatment. Increased % CD28^−^ of CD8^+^ T cells in patients with higher SLEDAI, treated with HCQ and cyclophosphamide or AZA, in comparison to those only on HCQ and with lower SLEDAI (*p* < 0.05).
Pavon E.J. et al., 2006 [56]	Cross-sectional	N = 51 SLE Age 38.1 (20–77) N = 36 HC Age 38.1 (20–77)	Increased % CD8^+^ T cells in SLE vs. HC (*p* < 0.0447). Decreased CD4/CD8 T cell ratio in SLE vs. HC (*p* < 0.0022). Increased CD38 expression on CD8^+^ T cells in SLE vs. HC (*p* < 0.002).	None reported.
Blanco P. et al., 2005 [30]	Longitudinal prospective	N = 61 SLE Mean: 35.5 (13–70) N = 36 HC Age not specified	No difference in % of CD3^+^CD8^+^ T cells in SLE vs. HC. Increased % HLA-DR^+^, perforin^+^ and granzyme B^+^ CD8^+^ T cells in active SLE vs. inactive SLE (*p* = 0.02, *p* < 10^−6^, *p* < 10^−6^, respectively) and HC (*p* < 10^−6^). Decreased % naïve CD8^+^ T cells in active SLE vs. inactive SLE (*p* < 10^−6^). Increased % TEMRA and EM CD8^+^ T cells in active SLE vs. HC and inactive SLE (*p* < 10^−6^).	Positive correlation between SLEDAI score and % granzyme B^+^ (r = 0.733, *p* < 10^−6^) and perforin^+^ CD8^+^ T cells (r = 0.731, *p* < 10^−6^).

^1^ Age in years expressed in this format, unless specified otherwise. Abbreviations used: ASM = age and sex matched, AZA = azathioprine, BILAG = British Isles Lupus Assessment Group Index, CI = confidence interval, CM = central memory, EM = effector memory, HC = healthy controls, HCQ = hydroxychloroquine, HLA-DR = human lymphocyte antigen-DR, IQR = interquartile range, LN = lupus nephritis, OR = odds ratio, SD = standard deviation, SLAMF7 = Signalling Lymphocytic Activation Molecule Family Member 7, SLE = systemic lupus erythematosus, SLEDAI = Systemic Lupus Erythematosus Disease Activity Index, TEMRA = terminally differentiated effector memory T cells.

The regulatory suppressor cell population, characterized by its CD8^+^CD28^−^ T cell phenotype, has also been examined extensively in SLE. The majority of studies focusing on this subpopulation have reported elevated frequencies of CD3^+^CD8^+^CD28^−^ T cells in peripheral blood in SLE [68,69] with some reporting a correlation between CD28^−^ T cell frequencies and systemic lupus erythematosus disease activity index (SLEDAI) and an association with presence of LN [68]. Employing a different gating strategy, Tulunay et al. showed a decrease in the proportion of CD28^−^ T cells within the CD8^+^ T cell population in SLE [50]. However, in keeping with other studies CD8^+^CD28^−^ T cell population frequencies were higher in patients with more aggressive immunosuppressive treatment and higher SLEDAI score. 

The cytotoxic phenotype of CD8^+^ T cells has also been investigated in SLE. Phenotypic analysis of frequencies of perforin^+^ and granzyme B^+^ populations in peripheral blood in SLE has shown that CD8^+^ T cells exhibit an increased cytotoxic phenotype, in particular in active patients, as granzyme B^+^ and perforin^+^ populations were elevated in SLE compared to HC [30,63,70] and correlated positively with SLEDAI score [30,70]. In addition, a highly cytotoxic subpopulation of CD8^+^ EM T cells, IL7Rα^low^ EM T cells, was also significantly elevated in SLE and the proportion of these cells in peripheral blood also correlated positively with SLEDAI [71]. However, the opposite phenotype has also been described in SLE with reduced perforin^+^, granzyme B^+^, and degranulation marker CD107a^+^ CD8^+^ T cell frequencies reported in SLE patients compared to HC [29,72].

#### 3.1.2. SS

Of 381 reports identified in our literature search 20 described differences in CD8^+^ T cell phenotype in SS, SSc, PM or DM. Ten of these studies are highlighted in Table 3 (see Appendix A for detailed description of the remaining reports). In patients with SS, no differences in CD8^+^ T cell frequency [73,74,75] or absolute count [74,76] have been reported compared to HC. However, activated HLA-DR^+^ CD8^+^ T cell frequencies and absolute cell counts were consistently shown to be elevated in SS patients vs. HC [76,77] and correlated positively with disease activity and positive anti-Ro/La and rheumatoid factor (RF) serology [76,78]. Increased frequencies of specific subtypes of CD8^+^ T cells, such as CD8^+^CD28^−^ [79] and CD8^+^CCR7^+^ [80] T cells, and decreased frequencies of late effector memory T [74] have also been reported in pSS, compared to HC. Frequencies of CD8^+^CD28^−^ T cells correlated negatively with symptoms such as dryness, fatigue, and pain as well as total EULAR Sjögren’s syndrome disease activity index (ESSDAI). However, decrease in frequency of CD8^+^CD28^−^ T cells also correlated with higher disease activity in the cutaneous and muscular domains in patients with pSS [79].

#### 3.1.3. SSc

There was no apparent consensus in the literature regarding CD8^+^ T cell peripheral blood frequencies in SSc in the 6 reports we identified. One study reported increased CD8^+^ T cell frequencies in SSc compared to HC, albeit this did not associate with age, treatment or extent of symptoms [81]. Others have described decreased CD8^+^ T cell percentages in dcSSc compared to HC with no differences in frequencies of CD8^+^ T cells in lcSSc vs. HC [82] or in absolute CD8^+^ T cell count in SSc vs. HC [83]. Another study reported decreased absolute CD8^+^ T cell counts in SSc vs. HC, though no differences in frequencies of CD8^+^ T cells were observed [84]. A large study involving 53 SSc patients and 33 HC reported differences in CD8^+^ T cell subpopulations in SSc vs. HC. Frequencies of CD8^+^ TEMRA cells were elevated in patients with SSc, while CM CD8^+^ T cells frequencies were diminished in the disease [85].

In addition, CD8^+^CD28^−^ T cells were increased in SSc compared to HC, and in dcSSc vs. lcSSc. This cell subset was found to be highly cytotoxic, as evidenced by high levels of IFN-γ expression upon phorbol 12-myristate 13-acetate (PMA) stimulation as well as ex vivo perforin and granzyme B expression relative to the CD28^+^CD8^+^ T cell subset [86]. The frequency of these cells correlated positively with fibrosis skin scores and with increasing age in patients as well as controls. Furthermore, in a regression analysis, CD8^+^CD28^−^ T cell frequency was a significant predictor of SSc diagnosis regardless of age [86].

**Table 3 ijms-23-11431-t003:** Altered peripheral blood CD8^+^ T cell phenotype in adults with connective tissue disease.

Author et al., Year [Ref]	Type of Study	N: Patients or Controls Age ^1^: Mean ± SD or Median (Range) or [IQR]	Altered CD8^+^ T Cell Phenotype	Clinical Relevance
**Sjögren’s syndrome (SS)**				
Martin-Gutierrez L. et al., 2021 [73]	Longitudinal clinical data, cross-sectional phenotyping	N = 45 pSS Mean: 59 (30–78) N = 29 SLE Mean: 48 (21–72) N = 14 SLE/SS Mean: 55 (26–56) N = 31 HC Mean: 44 (20–77)	No differences in % CD8^+^ T cell populations in pSS vs. HC when adjusted for age and ethnicity.	Frequencies of CD8^+^ T cell subpopulations could be used to stratify patients with pSS, SLE, and SLE/SS and predict long-term disease activity and damage trajectories in those with low or no disease activity.
Tasaki S. et al., 2017 [78]	Cross-sectional	N = 30 pSS 39.33 ± 9.44 N = 30 HC 61.07 ± 10.8	Transcriptomic and proteomic differences in CD8^+^ T cells in pSS vs. HC used to derive a unique pSS disease signature.	Positive correlation between %CD8^+^ T cell TEMRA and levels of anti-Ro, anti-La antibodies and IgA. Positive correlation between % HLA-DR^+^ CD8^+^ T cells and anti-Ro antibody levels.
Narkeviciute I. et al., 2016 [74]	Cross-sectional	N = 30 pSS Mean: 57 (32–78) N = 14 nSS Mean: 59 (44–91)	No difference in % or absolute count of total CD8^+^ T cells in pSS vs. nSS. Increased % memory CD8^+^CD57^+^CD27^+^CD45RA^−^ T cells in pSS vs. nSS (*p* = 0.0028). Decreased % and absolute counts of cytotoxic effector CD8^+^CD57^+^CD27^−^CD45RA^+^ T cells and TEMRA CD8^+^CD57^−^CD27^−^CD45RA^+^ T cells in pSS vs. nSS (*p* = 0.0184 and *p* = 0.0436, respectively).	Negative correlation between % CD8^+^CD57^−^CD27^+^CD45RA^−^ memory T cells and Schirmer’s I test (r = −0.429, *p* = 0.029) in pSS.
Mingueneau M. et al., 2016 [76]	Cross-sectional	N = 49 pSS 54 [43.5–63.5] N = 45 HC or nSS 53 [32.5–62]	No difference in absolute CD8^+^ T cell count in SS vs. HC. Increased % of activated (HLA-DR^+^) CD8^+^ T cells in pSS vs. HC (*p* = 0.0007).	Positive correlation between disease activity and % activated HLA-DR^+^ CD8^+^ T cells (r = 0.51, *p* = 0.007). Positive correlation between % activated CD8^+^ T cells and serum anti-Ro/La antibodies (r = 0.57, *p* = 0.002) and rheumatoid factor (r = 0.43, *p* = 0.027).
Smolenska Z. et al., 2012 [79]	Longitudinal prospective	N = 16 pSS 50 [39–60] N = 7 sSS 55 [45–70] N = 10 HC 48 [44–67]	Increased % CD8^+^ CD28^−^ T cells in pSS vs. HC (*p* = 0.01) and sSS (*p* < 0.02). No difference in absolute number of CD8^+^CD28^−^ T cells.	Negative correlation between % CD8^+^CD28^−^ T cells and dryness/fatigue/pain in pSS (r = −0.44, −0.58, −0.71, *p* < 0.05 for all). Negative correlation between % CD8^+^CD28^−^ T cells and total ESSDAI (r = −0.43, *p* < 0.05). Decrease in % CD8^+^CD28^−^ T cells correlated with higher disease activity in the cutaneous (*p* = 0.04) and muscular (*p* = 0.02) ESSDAI domains.
**Systemic scleroderma (SSc)**				
Li G. et al., 2017 [86]	Cross-sectional	N = 65 SSc 49.1 ± 15.3 N = 35 HC (20–72)	Increased % CD8^+^CD28^−^ T cells in SSc vs. HC (*p* < 0.0001). Increased % CD8^+^CD28^−^ T cells in dcSSc vs. lcSSc (*p* < 0.0001). % CD8^+^CD28^−^ T cells correlated with increasing age in SSc and HC (r = −0.51, *p* < 0.0001). % CD8^+^CD28^−^ T cells significant predictor of SSc presence regardless of age (*p* = 0.04).	Positive correlation between % CD8^+^CD28^−^ T cells and Rodnan fibrosis skin scores after adjusting for age (r = 0.72, *p* < 0.001).
Fuschiotti P. et al., 2009 [85]	Cross-sectional	N = 53 SSc (22 lcSSc, 31 dcSSc) 48.6 ± 12.2 N = 33 HC, ASM	Increased % CD8^+^ TEMRA (CD45RA^+^CD27^−^) in SSc vs. HC (*p* < 0.01). Decreased % CM CD8^+^ (CD45RA-CD27^+^) T cells in SSc vs. HC (*p* < 0.01).	None reported.
**Polymyositis and Dermatomyositis (PM/DM)**				
Wang D.X. et al., 2012 [87]	Longitudinal prospective	N = 19 PM 48.00 ± 14.21 N = 70 DM 50.96 ± 14.01 N = 60 HC 47.63 ± 8.45	Decreased absolute CD8^+^ T cell counts in active DM vs. HC (*p* < 0.05). No difference in % CD8^+^ T cells.	Positive correlation between low CD3^+^CD8^+^ T cell count in PM/DM and MYOACT-total disease activity score (*p* = 0.008) and immunosuppressive drug treatment (*p* = 0.034). CD3^+^CD8^+^ T cells count is an independent risk factor for death in PM/DM (*p* < 0.05).
Fasth A.E. et al., 2009 [88]	Cross-sectional	N = 40 PM 61 (24–79) N = 24 DM 55 (28–74) N = 41 HC 52 (28–82)	Increased % CD8^+^CD28^−^ T cells in PM vs. HC (*p* = 0.016). No significant difference in DM vs. HC; 98% of CD8^+^CD28^−^ T cells contained perforin.	% CD8^+^CD28^−^ T cells decreased with disease duration in PM/DM, partly compensated by increase in % CD8^+^CD28^−^ T cells with age (*p* = 0.0184).
Aleksza M. et al., 2005 [89]	Cross-sectional	N = 50 (13 active) PM 45.9 ±13.7 N = 49 (29 active) DM 46.9 ± 13.5 N = 32 HC 30.9 ± 9.1	Decreased % CD8^+^ T cells in active DM vs. HC (*p* < 0.01) and inactive DM (*p* < 0.05). No difference in PM vs. HC. Increased % activated HLA-DR^+^ CD3^+^ T cells in PM, DM vs. HC, regardless of disease activity (*p* < 0.01). Decreased % IFN-γ^+^CD8^+^ T cells in active DM vs. HC and non-active DM (*p* < 0.01).	None reported.

^1^ Age in years expressed in this format, unless specified otherwise. Abbreviations used: ASM=age and sex matched, CM = central memory, DM = dermatomyositis, dcSSc = diffuse cutaneous systemic sclerosis, EM = effector memory, ESSDAI = EULAR Sjögren’s syndrome disease activity index, HC = healthy controls, HLA-DR = human leukocyte antigen -DR, IQR = interquartile range, lcSSc = limited cutaneous systemic sclerosis, MYOACT = Myositis Disease Activity Assessment Visual Analogue Scales, nSS = non-autoimmune sicca syndrome, PM = polymyositis, pSS = primary Sjögren’s syndrome, sSS = secondary Sjögren’s syndrome, SD = standard deviation, SS = Sjögren’s syndrome, SLE = systemic lupus erythematosus, SSc = systemic scleroderma, TEMRA = terminally differentiated effector memory T cells.

#### 3.1.4. PM and DM

Five reports described the CD8^+^ T cell peripheral blood phenotype in PM and DM. A large study involving 50 PM patients, 49 DM patients and 32 HC revealed decreased frequencies of CD8^+^ T cells in patients with active DM compared to HC and compared to DM patients with inactive disease [89]. Others also reported decreased absolute CD8^+^ T cell counts in DM compared to HC and low CD3^+^CD8^+^ T cell counts were associated with higher disease activity scores in PM and DM [87]. Increased frequencies of activated HLA-DR^+^ CD3^+^ T cells in PM and DM compared to HC, irrespective of disease activity were observed and frequencies of CD8^+^ T cells expressing the effector cytokine IFN-γ in active DM were found to be diminished compared to inactive DM and HC [89]. This reduction was not observed in active or inactive PM [89]. Others have reported reduced CD8^+^ CM T cell frequencies in patients with PM compared to HC [90] and perturbations in the CD8^+^ T cell repertoire in PM vs. DM and PM vs. HC due to oligoclonal CD8^+^ T cell expansion [91]. Finally, in keeping with findings in other CTDs, CD8^+^CD28^−^ T cell frequencies were increased in PM vs. HC. There was also a trend for an increase in DM, though this did not reach statistical significance [88]. 

### 3.2. Altered CD8^+^ T Cell Phenotype in Children and Adolescents with CTD

Very few human studies have focused on the role for CD8^+^ T cells in juvenile CTDs. Our literature search identified only 8 such reports (Table 4), 5 of which explored the CD8^+^ T cell phenotype in JSLE and the remaining 3 in JDM. In JSLE, as was the case with aSLE, the findings were at times conflicting. 

#### 3.2.1. JSLE

Two studies involving 60 or more JSLE patients reported an increase in total CD8^+^ T cell frequency in JSLE compared to HC [92,93], one of which, also reported reduced frequencies of CM, EM, and TEMRA CD8^+^ T cells in peripheral blood of patients with JSLE [93]. Lerkvaleekul et al. found the increase in total CD8^+^ T cell frequency in JSLE to be associated with mild disease and absence of vasculitis and LN [92]. Robinson et al., on the other hand, reported that JSLE patients with elevated CD8^+^ EM cells had higher disease activity over time, increased use of mycophenolate mofetil (MMF), and increased prevalence of LN, highlighting a possible role for these cells in the pathology of more severe JSLE [93]. Others have reported no difference in frequencies of total CD8^+^ populations or CD8^+^ T cell subpopulations in JSLE compared to HC [94,95]. In keeping with data from aSLE, absolute peripheral blood CD8^+^ cell counts were reduced in JSLE regardless of disease activity and the expression of activation marker CD38 was increased in JSLE patients compared to HC [95].

Recently, a transcriptomic analysis of single-cell RNAseq data from peripheral blood mononuclear cells (PBMCs) in JSLE has identified fractions of CD8^+^ T cells in JSLE expressing a strong cytotoxic program [96]. The authors confirmed their transcriptomic findings using flow cytometry in a limited cohort. Frequencies of granzyme B^+^ CD8^+^ T cells and perforin^+^ CD8^+^ T cells were increased in active JSLE patients compared to HC. 

#### 3.2.2. JDM

All 3 studies examining CD8^+^ T cell phenotypes in JDM reported a decrease in CD8^+^ T cell frequency [90,97] or absolute count [98] in the disease. Wilkinson et al. also observed a reduction in CD8^+^ CM cells in JDM compared to HC [90]. However, none of the studies reported any correlation between CD8^+^ T cell frequencies and clinical phenotype. We found no reports examining the role of CD8^+^ T cells in juvenile SS or juvenile SSc.

**Table 4 ijms-23-11431-t004:** Altered peripheral blood CD8^+^ T cell phenotype in children and adolescents with connective tissue diseases.

Author et al., Year [Ref]	Type of Study	N: Patients or Controls Age ^1^: Mean ± SD or Median (Range) or [IQR]	CD8^+^ T Cell Populations	Clinical Relevance
**Juvenile Systemic Lupus Erythematosus (JSLE)**				
Lerkvaleekul B. et al., 2021 [92]	Longitudinal prospective	N = 60 JSLE 12.15 [9.95–14.45] N = 42 HC (10–15)	Increased % total CD8^+^ T cells in JSLE vs. HC (*p* = 0.0015).	Increase in % total CD8^+^ T cells associated with absence of LN (*p* = 0.0017), absence of vasculitis (*p* = 0.0119), and inactive disease (*p* = 0.0034).
Robinson G. et al., 2020 [93]	Longitudinal clinical data, cross-sectional phenotyping	N = 67 JSLE 19 [13–25] N = 39 HC 18 [16–25]	Increased % total CD8^+^ T cells in JSLE vs. HC (*p* = 0.0006). Increased % CD8^+^ naïve T cells in JSLE vs. HC (*p* = 0.0005). Decreased % CD8^+^ CM (*p* = 0.0024), CD8^+^ EM (*p* = 0.016) and CD8^+^ TEMRA T cells (*p* = 0.038) in JSLE vs. HC.	Patients with elevated CD8^+^ and CD8^+^ EM T cells had more active disease over time, increased treatment with MMF and increased prevalence of LN.
Nehar-Belaid, D. et al., 2020 [96]	Cross-sectional	N = 33 JSLE Mean: 15.85 (10–19) N = 11 HC Mean: 12.27 (7–18)	scRNAseq clustering: CD8^+^ T cell subclusters expressing cytotoxic genes and IFN signature genes over-represented in JSLE vs. HC. Flow cytometry: 17 JSLE and 14 HC. Increased % granzyme B^+^ CD8^+^ cells in active (*p* = 0.029) and inactive JSLE (*p* = 0.035) vs. HC. Increased % perforin^+^ CD8^+^ T cells in active JSLE vs. HC (*p* = 0.044).	No association with disease severity or MMF use between patients stratified based on CD8^+^ T cell cluster.
Zahran A. et al., 2016 [94]	Longitudinal Interventional	N = 20 JSLE 9.4 ± 3.7 N = 20 HC 9.0 ± 4.5	No difference in % of CD8^+^ T cells in JSLE vs. HC. Decreased CD4/CD8 T cell ratio in JSLE vs. HC (*p* = 0.001). Decreased % of CD8^+^ Tregs (CD8^+^CD25^+^) (*p* < 0.001) and CD8^+^ Treg FoxP3 MFI (*p* = 0.016) in JSLE vs. HC.	Increase in CD8^+^ Tregs associated with decrease in SLEDAI (*p* = 0.01), CRP (*p* < 0.01), ESR (*p* = 0.004), protein in urine (*p* = 0.041), and improvement in clinical parameters (WBC (*p* = 0.018), C3 (*p* = 0.009), C4 (*p* = 0.014), albumin (*p* = 0.001), creatinine (*p* = 0.002)) upon royal jelly supplementation in JSLE.
Miyamoto M. et al., 2011 [95]	Cross-sectional	N = 30 JSLE 13 ± 2 N = 14 HC 14 ± 3	Decreased absolute CD8^+^ T cell counts in active (*p* = 0.01) and inactive (*p* = 0.02) JSLE vs. HC. No differences in % naïve, CM or EM CD8^+^ T cells in JSLE vs. HC. Decreased % TEMRA CD8^+^CD45RA^+^CCR7^−^ T cells in JSLE vs. HC (*p* = 0.01). Increased number of CD38 molecules on CD8^+^ T cells in JSLE patients vs. HC (*p* = 0.01). Decreased % CD8^+^CD28^+^ T cells in JSLE vs. HC (*p* < 0.05).	No association between changes in CD8^+^ T cell subsets and disease activity.
**Juvenile Dermatomyositis (JDM)**				
Wilkinson M. et al., 2020 [90]	Cross-sectional	N = 15 JDM 21.48 [19.07–23.19] N = 15 HC 20.11 [17.40–22.28]	Decreased % CD8^+^ CM T cells (*p* = 0.0202) and % total CD8^+^ T cells in JDM vs. HC (*p* = 0.0168).	None reported.
O’Gorman M.R. et al., 2000 [97]	Cross-sectional	N = 10 JDM 5.9 ± 0.9 N = 12 HC 7.9 ± 0.6	Decreased % CD8^+^ T cells in JDM vs. HC (*p* = 0.01).	None reported.
McDouall R.M. et al., 1990 [98]	Cross-sectional	N = 16 JDM (no age reported) N = 18 HC (no age reported)	Decreased absolute CD8^+^ T cell numbers in JDM vs. HC (*p* < 0.001).	None reported.

^1^ Age in years expressed in this format, unless specified otherwise. Abbreviations used: CM = central memory, C3 = complement 3, C4 = complement 4, CRP = C-reactive protein, EM = effector memory, ESR = erythrocyte sedimentation rate, HC = healthy controls, IFN = interferon, IQR = interquartile range, JDM = juvenile dermatomyositis, JSLE = juvenile systemic lupus erythematosus, MFI = mean fluorescence intensity, MMF = mycophenolate mofetil, scRNAseq = single cell RNA sequencing, SLEDAI = Systemic Lupus Erythematosus Disease Activity Index, TEMRA = terminally differentiated effector memory T cells, WBC = white blood cells.

### 3.3. Functional CD8^+^ T Cell Abnormalities in CTD across Age

In addition to phenotypic abnormalities, 41 reports described CD8^+^ T cell functional abnormalities in CTD across age (Table 5). The majority of these reports focused on aSLE (27 reports), and the remainder described functional CD8^+^ T cell defects in SSc (10 reports), JSLE (2 reports), pSS (1 report), and DM (1 report). Several reports have suggested increased functional cytotoxicity in SLE, in particular in active disease. CD8^+^ T cells from patients with SLE have been shown to have an enhanced ability to kill target cells in vitro [30] as well as increased IFN-γ intracellular expression in response to PMA or CD28 stimulation [99,100]. Increased functional cytotoxicity was also observed in a CD226^high^ subset of CD8^+^ T cells in SSc [101]. CD226^high^CD8^+^ T cells from SSc patients displayed a higher capacity to kill human umbilical vascular endothelial cells (HUVECs) in in vitro assays compared to cells from HC and were elevated in patients with more severe skin sclerosis and extensive interstitial lung disease (ILD) compared to those with milder disease [101]. 

In contrast, others have shown decreased cytotoxic capacity in CD8^+^ T cells in SLE, as evidenced by reduced cytotoxic molecule expression and degranulation capacity [72,102] as well as diminished effector function in response to viral antigens [29,103], decreased IFN-γ intracellular expression in response to PMA stimulation [104,105], and decreased non-restricted cytolytic activity in vitro [106,107,108]. 

Impaired Epstein–Barr Virus (EBV)-specific CD8^+^ T cell responses have also been reported in SLE [109,110,111]. EBV-specific CD8^+^ T cells in patients with SLE exhibit signs of exhaustion; these cells are less cytotoxic and fewer EBV-specific CD8^+^ T cells can produce multiple cytokines in SLE than in HC. Elevated memory CD8^+^ T cells expressing exhaustion/activation markers programmed cell death protein 1 (PD-1) and T-cell immunoglobulin mucin-3 (Tim-3) have also been reported in patients with active and inactive SLE [49] and in patients with high IFN gene scores [112]. Surprisingly, transcriptomic analysis of CD8^+^ T cells purified from patients with SLE revealed that although CD8^+^ T cell exhaustion was associated with poor infection clearance, it predicted better disease prognosis [113]. This raises the possibility of induction of CD8^+^ T cell exhaustion as a possible therapeutic approach in SLE, but also highlights the potential consequences this approach may have for infection control in these patients.

Metabolic defects in CD8^+^ T cells have been reported in SLE and these abnormalities have been attributed to prolonged Type I IFN exposure and TCR stimulation which alters the function and morphology of mitochondria in CD8^+^ T cells [114]. Furthermore, the expression of genes involved in DNA damage response pathways and mitochondria-induced apoptosis in CD8^+^ T cells correlated with disease activity in SLE [114]. Enhanced apoptosis of CD8^+^ T cells and increased expression of pro-apoptotic molecules such as Fas, FasL, and caspase-3 have also been reported in adult-onset SLE [51,69,115] and JSLE [95,116] along with defects in costimulatory [117] and activation [118] pathways and increased expression of co-stimulatory (CD40L, CD86) [119,120] molecules. Enhanced CD8^+^ T cell apoptosis has also been reported in SSc [121]. 

Several papers have described increased IL-13 production by CD8^+^ T cells in SSc. Among the CTDs we focused on in this review, this feature was unique to SSc. Highly cytotoxic CD8^+^ T cell subsets such as CD226^high^ CD8^+^ cells, CD8^+^CD28^−^ cells and CD8^+^ effector T cells have been shown to express high levels of this pro-fibrotic cytokine [85,86,101]. The upregulation of IL-13 is thought to occur via decreased translocation of Tbet to the nucleus in SSc CD8^+^ T cells. Reduced Tbet binding to the transcription factor GATA-3 has been shown to upregulate GATA-3-mediated IL-13 transcription in these cells [122]. GATA-3 upregulation in CD8^+^ T cells has been identified as a biomarker of immune dysfunction in SSc, resulting in excessive IL-13 production [123]. Frequencies of IL-13 producing CD8^+^ T cells correlated very closely with frequencies of GATA-3 expressing cells and the frequencies of GATA-3 expressing cells, in turn, correlated positively with extent of skin thickening (modified Rodnan skin thickness score) in SSc. Elevated frequencies of GATA-3 expressing cells were also associated with presence of ILD and early disease [123].

**Table 5 ijms-23-11431-t005:** CD8^+^ T cell functional abnormalities in connective tissue diseases.

Type of CD8^+^T Cell Functional Abnormality	Type of Disease [Ref]	Clinical Correlation
Increased functional cytotoxicity/cytotoxic capacity	SLE [30,99,100], SSc [101]	Associated with active disease in SLE and disease severity in SSc.
Decreased functional cytotoxicity/cytotoxic capacity	SLE [29,102,103,104,105,106,107,108]	Expanded population of CD8^+^CD38^high^ T cells (with reduced cytotoxicity) in patients with increased rates of infections. Decreased intracellular IFN-γ expression in CD8^+^ T cells of SLE patients with active disease.
Impaired EBV-specific CD8^+^ T cell responses	SLE [109,110,111]	Both active and inactive patients have increased EBV viral load compared to HC and impaired EBV specific CD8^+^ T cell responses.
CD8^+^ T cell exhaustion/activation	SLE [49,112,113], AAV [113]	Associated with better SLE clinical outcome (% patients with flare-free survival). No association with disease activity.
Metabolic CD8^+^ T cell disfunction	SLE [114]	Genes belonging to mitochondria-induced apoptosis and DNA damage response pathways correlated with disease activity.
Enhanced CD8^+^ T cell apoptosis and expression of pro-apoptotic proteins	SLE [51,69,115], JSLE [95,116], SSc [121]	Enhanced apoptosis of CD8^+^ T cells in active SLE.
Dysregulation of costimulatory and activation pathways	SLE [117,118,119,120]	Arthritis and low CH50 more common in patients with CD86 expression on CD8^+^ T cells. Increased % CD40L^+^CD8^+^ T cells in both active patients and in remission. Higher absolute CD40L^+^CD8^+^ T cell numbers in active SLE.
Upregulated pro-fibrotic IL-13 production	SSc [85,86,101,122,123]	Associated with higher levels of skin fibrosis, early disease, and ILD.
Increased production of type 2 cytokines	SLE [124], SSc [125,126,127]	Associated with higher risk of progressive lung fibrosis and decline in lung capacity in SSc. High levels of IL-4 and IL-13 associated positively with presence of Scl-70 or anti-centromere antibodies and negatively with glucocorticoid treatment in SSc. Associated with active disease in SLE.
Type 1 Interferon gene signature	SLE [128], SSc [129], pSS [78], PM [130], DM [130]	IGS correlated with disease activity scores, CD8^+^ TEMRA and HLA-DR^+^ CD8^+^ normalized T cell counts in pSS.

Abbreviations used: AAV = antineutrophil cytoplasmic antibody (ANCA)-associated vasculitis, CH50 = total haemolytic complement activity, DM = dermatomyositis, DNA = deoxyribonucleic acid, EBV = Epstein–Barr virus, HLA-DR = human leukocyte antigen-DR, IGS = interferon gene signature, IL = interleukin, ILD = interstitial lung disease, PM = polymyositis, pSS = primary Sjögren’s syndrome, SLE = systemic lupus erythematosus, SSc = systemic sclerosis, TEMRA = terminally differentiated effector memory T cells.

GATA-3 is a transcription regulator of Tc2 cell differentiation, and it regulates the expression of not only IL-13, but also of other type 2 signature cytokines such as IL-4 and IL-5. In addition to IL-13, the increased production of other type 2 cytokines has been reported in SSc and SLE. Increased production of IL-13 and IL-4 by memory CD8^+^ T cells in peripheral blood in SSc associated positively with presence of Scl-70 or anti-centromere antibodies and in the case of IL-4 also associated negatively with glucocorticoid treatment in a multiple regression analysis [126]. A similar memory CD8^+^ T cell Tc2 bias has also been shown in patients with active SLE. Memory CD8^+^ T cells from these patients selectively expressed high levels of IL-4 and IL-5 compared to HC [124].

Type 1 IFN Gene Signature (IGS) refers to the gene expression profile of cells produced upon exposure to type 1 interferon. The role of type 1 IFN in CTDs has been extensively reviewed elsewhere [131,132], however, as expected, IFN gene signatures have been reported in transcriptomic studies of CD8^+^ T cells in SLE [128], pSS [78], SSc [129], PM and DM [130]. In addition, IGS in whole blood correlated with disease activity scores, and CD8^+^ TEMRA and HLA-DR^+^ CD8^+^ normalized T cell counts in pSS [78].

### 3.4. Organ-Specific CD8^+^ T Cell Profiles

#### 3.4.1. SLE

Clinical disease manifestations in CTDs reflect the tissue damage caused by inflammation and the aberrant autoimmune response. Organ-specific CD8^+^ T cell signatures have been found in SLE, which highlight a possible role for these cells in mediating tissue damage in the disease. Our search has identified 16 papers examining organ-specific CD8^+^ T cell profiles in SLE (Table 6). Much of the evidence for CD8^+^ T cell involvement in tissue damage in SLE comes from kidney biopsies taken from patients with LN. A recent transcriptomic analysis has identified the presence of three clusters of CD8^+^ T cells in kidney tissue from patients with SLE [133]. One of these clusters represented CD8^+^ T cells expressing a high level of cytotoxic genes (perforin, granzyme B, and granulysin), the second cluster contained cells expressing granzyme K, and the third had features of resident memory T cells. All three clusters expressed low levels of exhaustion markers compared to the levels seen in peripheral blood in the same patients, suggesting that CD8^+^ T cell exhaustion may not occur in the kidney in LN. The increased presence of CD8^+^ Trm CD103^+^ cells in kidneys of patients with LN compared to normal renal tissue was also independently confirmed by others using immunofluorescence staining [134].

In addition, multiple histological studies have identified the presence of CD8^+^ T cells in kidney biopsy tissue [62,72,135,136,137] with several reporting the CD8^+^ T cell as the predominant periglomerular and interstitial infiltrating cells [62,135,136]. Renal CD8^+^ T cell infiltration also strongly correlated with SLEDAI score, renal disease activity, and renal function [136,137]. In a longitudinal study involving 197 LN patients, multivariate regression analysis revealed that high numbers of CD8^+^ tubulointerstitial infiltrating T cells were independently associated with end stage renal disease (ESRD) [137]. CD8^+^ T cells expressing the degranulation marker CD107a have also been found in kidney infiltrates and a positive correlation between the number of CD107a^+^ T cells and proteinuria has been reported [72].

No correlation was found between the number of kidney infiltrating CD8^+^ T cells and the number of CD8^+^ T cells in urine [62,135]. However, increased CD8^+^ T cell counts in urine were associated with active LN and correlated with SLEDAI disease activity [62,138]. In fact, it has been reported that CD8^+^ T cell counts in urine can be used to discriminate patients with active and inactive LN with 100% specificity and sensitivity [138]. In contrast, others have reported that CD4/CD8 T cell ratio in urine was increased in patients with LN, due to elevated numbers of CD4^+^ T cells in the urine compared to those without LN and correlated with SLEDAI disease activity [139].

Cutaneous manifestations are present in up to 85% of SLE patients [140]. Skin biopsies have revealed that CD8^+^ T cells are the dominant infiltrating cell in lupus skin lesions, in contrast to DM where CD4^+^ T cells predominate [141,142]. The cells infiltrating the dermis expressed granzyme B [142] and a relative increase in CD8^+^ lymphocytes has been reported in central lesions, suggesting that CD8^+^ T cells may be driving the expansion and persistence of these lesions [143]. CD8^+^ T cells have also been found in oral lesions in SLE, though CD4^+^ T cell predominate [144].

Phenotyping of bronchoalveolar lavage fluid (BALF) lymphocytes from patients with SLE indicated that there was no difference in frequency or absolute number of CD8^+^ T cells compared to HC [145]. However, HLA-DR expression was markedly increased in CD4^+^ and CD8^+^ T cells in BALF compared to blood, indicating local T cell activation in the lungs of patients with SLE. The authors also noted negative correlations between BALF CD8^+^ T and NK (natural killer) cell absolute counts and frequencies with parameters of pulmonary diffusing capacity; increasing CD8^+^ T cells and NK cells were associated with diminished lung function.

**Table 6 ijms-23-11431-t006:** Organ-specific CD8^+^ T cell profiles in SLE.

Organ Involvement	CD8^+^ T Cell Signatures [Ref]	Clinical Relevance
Lupus nephritis (kidney biopsy)	CD8^+^ T cell clusters identified in transcriptomic analysis of kidney tissue. No upregulation of CD8^+^ T cell exhaustion markers [133].	None reported.
	CD8^+^ T cells are the predominant kidney infiltrating cells [62,72,135,136,137].	Renal CD8^+^ T cell infiltration correlates with the renal activity index (r = 0.63, *p* = 0.0007) [136] high serum creatinine levels (r = 0.75, *p* = 0.0001) [136,137], SLEDAI (r = 0.14, *p* < 0.05) [137], proteinuria (r = 0.11, *p* < 0.05), glomerulosclerosis (r = 0.42, *p* < 0.001), and degree of tubulointerstitial inflammation (r = 0.46, *p* < 0.001) [137]. CD8^+^ T cell infiltrates associated with poor response to induction therapy [136] and ESRD progression (*p* < 0.001) [137].
	Elevated number of CD8^+^CD103^+^ Trm cells in LN kidney compared to healthy kidney tissue (*p* < 0.01) [134].	None reported.
Lupus nephritis (urine)	Elevated CD8^+^ T cell numbers in SLE with active renal disease vs. HC (*p* < 0.005) [62] and SLE without active renal disease (*p* < 0.001) [62], (*p* < 0.005) [135], (*p* < 0.0001) [138]. Nearly 70% of urinary CD8^+^T cells express the EM phenotype [62].	Increased CD8^+^T cell counts/mL in urine associated with active renal disease and correlated with SLEDAI (r = 0.68, *p* < 0.001) [62], (r = 0.7641, *p* < 0.0001) [138]. Urine CD8^+^ T cell counts used to discriminate patients with active LN and inactive LN and active LN vs. no renal involvement [135,138]. No correlation between urinary T-cell counts and renal activity index [135].
	T cells present in non-LN patients are predominantly CD8^+^ [139].	Positive correlation between urinary CD4/CD8 T cell ratio and SLEDAI (r = 0.38, *p* = 0.0047). Elevated CD4/CD8 T cell ratio associated with LN.
Cutaneous (skin biopsy)	CD8^+^ T cells are dominant infiltrating cells in majority of SLE patients [141], express granzyme B [142], and are elevated in central lesion sites vs. peripheral sites (*p* < 0.01) [143].	None reported.
Cutaneous (oral lesions)	CD8^+^ T cells present in oral lesions in SLE though CD4^+^ T cells predominate [144].	None reported.
Pulmonary (BALF)	No difference in % or absolute CD8^+^ T cell number or CD4/CD8 T cell ratio in BALF. Elevated % of CD8^+^HLA-DR^+^ T cells in BALF compared to blood [145].	No correlation with disease activity (Liang score). Tendencies for inverse correlations between % or number of CD8^+^ T cells with lung function parameters: transfer factor for carbon monoxide (r = −0.47, *p* = 0.07) and diffusing capacity of the alveolocapillary membrane (r = −0.47, *p* = 0.06).
Neuropsychiatric (PBMC)	IFN-γ secreting myelin-specific CD8^+^ T cells detected in peripheral blood in SLE with neuropsychiatric lupus without APS, but with white matter lesions [146].	None reported.

Abbreviations used: APS = antiphospholipid syndrome, BALF = bronchoalveolar lavage fluid, EM = effector memory, ESRD = end-stage renal disease, HC = healthy controls, HLA-DR = human leukocyte antigen- DR, IFN = interferon, LN = lupus nephritis, PBMC = peripheral blood mononuclear cells, SLE = systemic lupus erythematosus, SLEDAI = Systemic Lupus Erythematosus Disease Activity Index.CD8^+^ T and NK (natural killer) cell absolute counts and frequencies with parameters of pulmonary diffusing capacity; increasing CD8^+^ T cells and NK cells were associated with diminished lung function.

Finally, in neuropsychiatric lupus IFN-γ secreting myelin-specific CD8^+^ T cells were detected in peripheral blood of a subset of SLE patients. They were exclusively found in SLE patients with white matter lesions without antiphospholipid syndrome, suggesting a possible role for these cells in tissue damage in neuropsychiatric SLE [146]. Taken together this data indicates that CD8^+^ T cells may be important in mediating specific organ damage in SLE and may also play a role in perpetuating the autoimmune response.

#### 3.4.2. SS

Evidence for organ-specific involvement of CD8^+^ T cells in other CTDs has also been reported. Our search identified 25 such reports: 6 in SS, 6 in SSc, 12 in PM/DM and 1 report including both patients with PM/DM and SSc (Table 7). Data from lacrimal and salivary gland biopsies from patients with SS has revealed the presence of CD8^+^ T cell infiltration at these sites [76,147]. Fujihara et al. reported the presence of CD8^+^ T cells (but not CD4^+^ T cells) around the acinar epithelial cells in patients with SS compared to non-SS [147]. The majority of these CD8^+^ T cells expressed CD103^+^, thereby identifying them as CD8^+^ Trm cells. The authors also showed that acinar cells adhered to CD8^+^ T cells within the lacrimal biopsy tissue were apoptotic and that the expression of cytotoxic death mediators granzyme B and perforin as well as Fas and FasL were significantly increased in biopsy tissues from SS patients compared to non-SS [147,148]. 

More recently, Mingueneau et al., confirmed increased CD8^+^ T cell numbers and elevated frequencies of activated HLA-DR^+^ CD8^+^ T cells in labial biopsies in SS compared to non-SS and frequencies of activated CD8^+^ T cells in the lip biopsy correlated positively with ESSDAI scores [76]. In contrast, however, an investigation by Ohlsson et al. found that although ductal and acinar CD8^+^ T cell infiltration was present in pSS and sSS labial biopsies, it was not statistically significant compared to non-SS controls and apoptosis of epithelial cells was very rare (less than 1%) in SS [149]. 

T cells in the labial glands of SS patients have also been shown to exhibit c-Jun N-terminal kinase (JNK) pathway activation [150] and to express B cell activating factor (BAFF) [151], though these findings were not specific to CD8^+^ T cells, as they also occurred in CD4^+^ infiltrating T cells, and their significance is at present unclear.

#### 3.4.3. SSc

In SSc, CD8^+^ T cells were the predominant infiltrating cell type in the perimysium in muscle biopsies [152] and elevated infiltrating CD8^+^ T cells compared to CD4^+^ T cells were found in skin biopsies in early stage SSc [153]. However, this reversed in late stage biopsies [153] and no detectable clonal CD8^+^ T cell expansion in the skin was seen in long-term SSc [82]. CD8^+^CD28^−^ T cells were present in early stage dcSSc and these cells exhibited features of Trm as evidenced by increased CD69 expression [86]. Elevated CD8^+^ T cell counts were also reported in BALF in patients with SSc compared to HC [83,125] and the number of CD8^+^ T cells in BALF correlated with parameters of pulmonary function [83]. Furthermore, CD8^+^ BALF cells in SSc patients expressed the type 2 cytokine transcriptional profile [125,127] and patients with this profile had higher risk of progressive lung disease [127].

#### 3.4.4. PM and DM

CD8^+^ T cells have also been detected in muscle and vessel infiltrates in patients with PM, and to a lesser extent in DM [88,152,154,155,156]. In particular, elevated numbers of CD8^+^ T cells were present in the endomysium in PM compared to DM [152,154] and CD8^+^ T cells were the predominant cell type infiltrating non-necrotic muscle fibres in PM [152]. Detailed phenotyping of perivascular and endomysial CD8^+^ T cell infiltrates in PM and DM identified them to be predominantly memory (CD45RO^+^) cells [155,157], while infiltrates in PM were also shown to express cytotoxic effector molecules perforin and granzyme B [158,159,160,161,162]. The presence of these highly cytotoxic CD8^+^ T cell populations in muscle fibres in PM, suggests a possible role for these cells in mediating tissue damage at these sites. In support of this, CD8^+^ T cells and granzyme B^+^ T cells have been found to cluster around apoptotic myonuclei in muscle fibres in PM [160] and in in vitro experiments CD8^+^CD28^−^ cells from PM patients have been shown to be myotoxic in a perforin-dependent manner toward autologous skeletal muscle cells [161].

**Table 7 ijms-23-11431-t007:** Organ-specific CD8^+^ T cell profiles in other CTDs.

Organ Involvement	CD8^+^ T Cell Signatures [Ref]	Clinical Relevance
Sjögren’s syndrome (salivary gland biopsy)	Increased CD8^+^ T cell counts (*p* = 0.007) and frequency of activated CD8^+^HLA-DR^+^ T cells in pSS vs. non pSS in labial gland biopsy (*p* = 0.0097) [76].	None reported.
	CD8^+^ T cells are localized to acinar epithelial cells in lacrimal and salivary glands and express integrin CD103, which facilitates epithelial cell apoptosis via Fas expressed on acinar epithelial cells and perforin/granzyme cytotoxicity [147,148].	None reported.
	No difference in acinar or ductal CD8^+^ T cell counts in pSS or sSS compared to HC, few apoptotic cells present in tissue sections [149].	None reported.
	CD8^+^ infiltrating labial T cells express BAFF [151] and exhibit JNK cascade activation [150].	None reported.
Systemic sclerosis (muscle biopsy)	CD8^+^ T cells present in perivascular and perimysial sites and are predominant infiltrating cell type in perimysium [152].	None reported.
Systemic sclerosis (skin biopsy)	CD8^+^ T cells in skin of patients with early dcSSc are mostly CD28-(72.3 ± 13.8%). Skin CD8^+^CD28^−^ express Trm marker CD69. Few cells express CD103 [86].	None reported.
	Elevated infiltrating CD8^+^ vs. CD4^+^ T cell numbers in early-stage systemic sclerosis (*p* < 0.0001). This is reversed in late-stage biopsies. No CD8^+^ T cells found in normal skin [153].	None reported.
	No detectable CD8^+^, CD4^+^ or total T cell clonal expansion in long-standing SSc [82].	None reported.
Systemic sclerosis (BALF)	Elevated CD8^+^ T cell count in SSc vs. HC (*p* < 0.05) [83], (*p* < 0.01) [125]. CD8^+^ T cells in BALF of SSc made type 2 cytokine mRNA (IL-4 and or IL-5) while HC did not [125].	Number of CD8^+^ T cells in BALF correlates with FVC (forced vital capacity) (r = 0.4, *p* < 0.05) [83] Patients whose BAL cells made type 2 cytokine mRNA had decreased FVC over time post BAL (*p* = 0.04) [125].
	Transcriptomic analysis of BALF CD8^+^ T cells identified a subset of SSc patients with CD8^+^ T cell activation, a type 2 cytokine phenotype, reduced activation-induced cell death, and production of profibrotic factors [127].	Patients in this subset had higher risk of progressive lung disease.
Dermatomyositis (skin biopsy)	CD8^+^ T cells present in infiltrates and their distribution across muscle sites is similar in ADM and JDM [163].	No correlation with clinical parameters.
Myositis (muscle biopsy)	CD8^+^ T cells present in muscle and vessel infiltrates in DM and PM. CD8^+^ T cell numbers vary in DM across muscle connective tissue sites. Elevated number of CD8^+^ T cells in endomysium in PM vs. DM [154].	None reported.
	CD8^+^ T cells were predominant cell type infiltrating non-necrotic muscle fibres in PM [152] and are present in endomysial, perivascular and perimysial sites in PM in greater number than CD4^+^ T cells. [156] Positive gradient for % of CD8^+^ T cells in DM and PM between perivascular and endomysial sites. Increased % of CD8^+^ T cells and % activated T cells in endomysium in PM vs. DM [152].	None reported.
	Perivascular CD8^+^ T cells in PM and DM [155] and endomysial CD8^+^ T cells in PM are predominantly CD45RO^+^ [157]. Elevated % CD8^+^CD45RO^+^ T cells in muscle biopsies compared to peripheral blood of HC. T cells surrounding invaded muscle fibres were mostly CD8^+^ [155].	No correlation with age, duration of illness, or serum CK [157].
	CD8^+^, Granzyme B^+^, and perforin^+^ T cells predominate in endomysium in PM. Rare in endomysium in DM. [158,159,162]. CD8^+^ T cells and Granzyme B^+^ T cells cluster around apoptotic myonuclei in muscle fibres in PM [160].	None reported.
	Infiltrating CD8^+^ T cells in PM and DM are predominantly CD8^+^CD28^−^. [88] CD8^+^CD28^−^ T cell myotoxicity in PM is mediated via perforin, granzyme B and IFN-γ [161].	Positive correlation between % CD8^+^CD28^−^ T cells and global disease activity (r = 0.90, *p* = 0.01) [88].
Myositis (lung biopsy)	CD8^+^ T cells diffusely distributed across lung biopsies in DM and PM patients with interstitial pneumonia. CD8^+^ T cells predominate over CD4^+^ T cells. No difference between DM and PM [164].	None reported.

Abbreviations used: ADM = adult dermatomyositis, BAFF = B-cell activating factor, BAL = bronchoalveolar lavage, BALF = bronchoalveolar lavage fluid, DM = dermatomyositis, FVC = forced vital capacity, HC = healthy controls, HLA-DR = human leukocyte antigen- DR, IFN = interferon, IL = interleukin, JDM = juvenile dermatomyositis, JNK = c-Jun N-terminal kinase, mRNA = messenger ribonucleic acid, PM = polymyositis, pSS = primary Sjögren’s syndrome, sSS = secondary Sjögren’s syndrome, SSc = systemic sclerosis.

In addition to muscle inflammation, DM is also characterized by inflammation in the skin. Skin biopsies in DM patients have shown the presence of CD8^+^ T cell infiltrates, however their distribution and number were similar to CD4^+^ T cells and no differences in CD8^+^ T cell number or distribution were found between JDM and ADM [163]. Furthermore, CD8^+^ T cells have been detected in higher number than CD4^+^ T cells in lung biopsies of PM/DM patients with interstitial pneumonia [164].

## 4. Discussion

### 4.1. Commonalities in CD8^+^ T Cells and Subpopulations in Peripheral Blood across Adult CTDs

In this comprehensive review we have examined in detail recently published scientific literature concerning CD8^+^ T cell peripheral blood phenotype, function, and organ-specific profiles in adult and juvenile CTDs. Despite many contradictory reports, in particular in aSLE, there were some commonalities in peripheral blood CD8^+^ T cell phenotypes across adult CTDs (summarized in Figure 2).

CD8^+^ T cell frequencies in aSLE were either increased [53,54,55,56,57,58,59] or unchanged [29,30,49,50,51,52] relative to HC and this was also the case in SSc [81,82,84]. Similarly, there were no differences in CD8^+^ T cell frequencies in SS [73,74,75] or PM [89,90] compared to HC. Notably, few significant differences in CD8^+^ T cell phenotype and immunophenotype in general were observed between SLE and SS, suggesting a shared disease pathology between SS and SLE [73]. In DM, CD8^+^ T cell frequencies were selectively decreased only in active DM patients compared to inactive DM and controls [89]. Thus, it is possible that the discrepancies in results in aSLE and SSc among different studies are due, at least in part, to varying disease activity states of the patient cohorts and possibly confounded by various medications used. Phenotyping of CD8^+^ T cell memory and effector T cell subtypes proved equally contentious in aSLE, though several studies suggested a decrease in CM CD8^+^ T cell frequency in aSLE compared to HC [29,61] in line with findings in SSc [85] and PM [90]. The reduction in memory cells in circulation in CTDs may indicate migration of these cells into tissues where they can mediate tissue damage and perpetuate the autoimmune response.

### 4.2. Shared CD8^+^ T Cell Activation and Cytotoxic Cell Profiles

T cell activation in autoimmune diseases can drive B cell activation, cytokine secretion and antibody production, thereby perpetuating disease in susceptible individuals. HLA-DR is a glycoprotein encoded by the HLA-DR MHC region and it is normally absent on resting cells [165]. Its expression on the CD8^+^ T cell surface is a marker of T cell activation [166]. Frequencies of activated HLA-DR^+^ CD8^+^ T cells were consistently increased in the blood across reports in all the examined CTDs [30,49,52,65,67,76,77,89], with the exception of SSc where no differences compared to HC were observed [82]. However, it should be noted that in all studies, increased frequencies of HLA-DR^+^CD4^+^ T cells were also observed in CTD vs. HC, indicating that T cell activation is not limited to the CD8^+^ T cell compartment in these autoimmune conditions. 

Another common feature of CD8^+^ T cells in peripheral blood in CTDs pertains to their reduced capacity to produce IFN-γ. Cytotoxic T cells produce IFN-γ in response to activation, and CD8^+^ T cell production of IFN-γ upon PMA stimulation has been shown by some to be diminished in SLE, in particular in active disease [104,105], though others have reported the opposite effect [99,100]. Reduced CD8^+^ T cell IFN-γ expression has also been described in patients with active DM [89]. As this was not the case in PM, this finding may support the notion that the pathologies of DM and PM are distinct.

### 4.3. CD8^+^CD28^−^ T Cell Phenotype across CTDs

In addition to similarities in cytotoxic CD8^+^ T cell subsets, some phenotypic similarities in the putative suppressive CD8^+^ Treg CD8^+^CD28^−^ T cell population also exist across the CTDs we examined. CD8^+^CD28^−^ T cells have been described as both immunosuppressive and cytotoxic. Irreversible loss of CD28 molecule surface expression in T lymphocytes is thought to occur as a result of chronic antigen exposure. This leads to accumulation of senescent, highly antigen experienced CD8^+^CD28^−^ cells [167]. Our review of the literature revealed that frequencies of CD8^+^CD28^−^ T cells were elevated in the blood of patients with SLE [68,69], SS [79], SSc [86], PM [88] and DM compared to HC [88]. Importantly, in all CTDs CD8^+^CD28^−^ were shown to be cytotoxic, rather than immunosuppressive. In SLE this was evidenced by lack of FoxP3 expression on CD28^−^ cells [68], their high production of the pro-inflammatory cytokine IFN-γ [69] and low expression of the immunosuppressive cytokine IL-10 [50,69]. In SS, decreased proportions of these cells in peripheral blood were associated with higher disease activity in the muscular and cutaneous ESSDAI domains, leading to speculations that T cells from the periphery may be migrating into local areas of disease, promoting tissue damage at those sites via local cytotoxicity [79]. Others have shown that although CD8^+^CD28^−^ T cells from patients with SS cultured for 72 h, show a trend for higher TGFβ expression than cells from HC, they also express significantly lower levels of IL-10 than cells from HC [168], further calling into question the immunosuppressive capacity of CD8^+^CD28^−^ T cells in SS.

In SSc CD8^+^CD28^−^ cells both in peripheral blood and in skin were found to express the effector/EM phenotype and expressed high levels of perforin and granzyme B [86] while in PM and DM CD8^+^CD28^−^ cells produced more pro-inflammatory TNFα than CD28^+^ cells on stimulation with anti-CD3 and 98% of CD8^+^CD28^−^ cells expressed perforin [88]. Thus, frequencies of cytotoxic CD8^+^CD28^−^ T cells in the blood are increased across CTDs, raising the possibility that these cells may be responsible for perpetuating autoimmunity and mediating tissue damage upon migration to local disease sites. 

### 4.4. Impact of Age on CD8^+^ T Cell Phenotype

Loss of CD28 expression is associated with CD8^+^ T cell senescence, thus unsurprisingly, frequencies of CD8^+^CD28^−^ T cells in the blood increase with age in healthy subjects [169]. As such, age is a potential confounder in interpreting CD8^+^CD28^−^ T cell phenotyping data and must be considered during analysis. Increased frequencies of CD8^+^CD28^−^ T cells were reported irrespective of age in SSc [86] and PM/DM [88]. Although the patients and controls were age and sex matched in all studies, regression analyses to correct for age were not employed in the above mentioned SLE studies [50,68,69], which may in part account for the disparities between the findings in SLE.

In addition, age-dependent variations in other CD8^+^ T cell subpopulations as well as total CD8^+^ T cells are well documented in healthy subjects. Both absolute CD8^+^ T cell counts and frequencies of CD8^+^ T cells are known to decline with age [170,171,172], as is also the case with naïve CD8^+^ T cells [171,172]. This is accompanied by a concomitant increase in CM, EM, effector, and HLA-DR^+^CD8^+^ T cell frequency. Absolute cell counts of these populations, however remained stable throughout [171]. Thus, age is a key factor to account for in CD8^+^ T cell phenotyping analyses, in particular when proportions of CD8^+^ T cells rather than absolute counts are reported. It is also of particular importance when comparing CD8^+^ T cell profiles between juvenile and adult-onset diseases.

### 4.5. Absence of Common CD8^+^ T Cell Peripheral Blood Immunophenotype across Juvenile CTDs

Our literature search identified only 8 studies focused on CD8^+^ T cell peripheral blood profiles in juvenile CTDs (5 in JSLE and 3 in JDM). Much like in adult disease, in JSLE CD8^+^ T cell frequencies were either reported as increased or unchanged while absolute cell counts were decreased [92,93,94,95]. In JDM total CD8^+^ T cell frequencies or absolute counts were reduced compared to HC across all studies [90,97,98]. Based on these limited data, there are no obvious commonalities in CD8^+^ T cell peripheral blood phenotype between JSLE and JDM, however the paucity of studies in juvenile CTDs and the small cohort sizes, in particular in JDM, prevents us from drawing definitive conclusions. Furthermore, we did not identify any studies directly comparing CD8^+^ T cell profiles in aSLE and JSLE. However, Wilkinson et al. reported no difference in CD8^+^ T cell frequencies in DM compared to age matched HC, whereas in JDM there was a significant reduction compared to a separate set of age matched HC [90], indicating that unique immune signatures are associated with juvenile and adult disease. 

### 4.6. Reasons for Variability in Investigations of Peripheral Blood Phenotypes

We found there to be considerable variability in the data across the examined studies, especially in aSLE. The variability of the data in relation to CD8^+^ T cell subtype frequencies and clinical correlations may be attributed to several factors including: differences in lab methodology (gating strategy, surface marker selection), small patient numbers, inherent heterogeneity of the patient cohorts, differences in organ involvement and treatment, as well as differences in the way disease activity is measured. Studies involving absolute cell counts tended to be more consistent and the results were easier to interpret than population frequencies. Although most studies relied exclusively on reporting frequencies rather than absolute counts, population frequencies are dependent on each other, therefore frequency data reported in one subpopulation alone should be interpreted with caution. 

### 4.7. Commonalities in CD8^+^ T Cell Function across CTDs

Our literature search has identified several commonalities in aberrant CD8^+^ T cell function across CTDs, specifically in SLE and SSc. Increased functional cytotoxicity, enhanced CD8^+^ T cell apoptosis and increased memory CD8^+^ T cell type 2 pro inflammatory cytokine production have been shown to be associated with disease severity in both diseases [30,51,69,101,115,121,124,126]. The precise significance of these findings is unclear at present, however enhanced apoptosis may produce apoptotic bodies containing autoantigens and increased exposure to these antigens along with increased pro-inflammatory cytokine production may perpetuate the autoimmune response in SLE and SSc.

Apart from Type 1 IFN disease signatures, no commonalities in CD8^+^ T cell function have been found between SLE, SS, SSc and PM or DM, however the absence of such data does not preclude the possibility that shared functional abnormalities do exist. CD8^+^ T cell functional experiments involving human samples are methodologically notoriously difficult to perform and interpret due to the heterogeneity of the disease cohorts and to the fact that autoreactive CD8^+^ T cell specificities in these CTDs are at present unknown. In addition, negative data is often not published, therefore we cannot confirm the absence of functional CD8^+^ T cell aberrations in SS, PM, or DM. Extensive immunophenotyping of peripheral blood in patients with SLE and SS has shown these patients have remarkably similar immunoprofiles [73] and though this cannot be used to infer deficiencies in CD8^+^ T cell function, it raises the possibility that these diseases have shared immunopathology. However, as autoimmune activation and tissue damage occurs at local disease sites, changes in CD8^+^ T cell subsets in peripheral blood do not reflect the full picture of the disease state, and organ-specific immune profiles may be more informative.

### 4.8. Disease-Specific CD8^+^ T Cell Organ Profiles

Studies involving affected tissues in CTDs are extremely valuable as they can provide insight into immune cell activation and infiltration directly at sites of local disease. However, tissue samples from disease sites are often difficult to obtain due to the invasive procedures required. As such, studies describing CD8^+^ T cell profiles which we examined often relied on fewer than 10 patients, especially in rarer CTDs where access to patients and tissues is severely constrained. In addition, due to the invasive nature of the procedures, the collection of tissue samples is frequently limited to patients in whom a rapid diagnosis confirmation is necessary. This means that patients in these studies typically had greater disease activity than patients in investigations focusing exclusively on immunophenotype and function in peripheral blood. Thus, studies examining both peripheral blood phenotypes and organ-specific immunoprofiles in the same patients are particularly valuable, especially those that aim to correlate blood profiles with immunological activity in target organs.

Cytotoxic CD8^+^ T cell infiltrates have been found in many of the tissues affected in CTDs. In kidney biopsies from LN patients infiltrating CD8^+^ T cell numbers correlate positively with SLEDAI and kidney function parameters and can be used to predict renal prognosis [137] while CD8^+^ T cell counts in urine can be used to differentiate patients with active LN vs. those without renal involvement or with inactive LN [135,138]. CD8^+^ T cells are also the predominant infiltrating cell in SLE skin biopsies and increased frequencies of activated CD8^+^ T cells are also present in BALF compared to peripheral blood in patients with SLE [141,145].

In pSS there was an increased number of activated HLA-DR^+^CD8^+^ T cells in labial gland biopsies which mirrored the trend found in the peripheral blood, [76], suggesting a possible role for CD8^+^ T cell phenotypes in the blood as a marker of target organ disease activity.

CD8^+^ T cells were the predominant infiltrating cell type in muscle and skin biopsies in SSc and in non-necrotic muscle fibres in PM [152], as well as target organs in SSc and PM/DM [125,153,164]. Importantly, CD8^+^ T cells in many of these tissue infiltrates in CTDs were found to be cytotoxic [133,147,153,158,159,162]. This observation combined with the reduction in memory CD8^+^ T cell subsets in peripheral blood in SLE, SSc and PM points towards possible migration of these cells from the periphery into tissues where they can mediate local tissue destruction. In addition, CD8^+^ Trm cells have also been reported in target organs in SLE, SS, and SSc [86,134,147]. In contrast with other CTDs, CD8^+^ T cells were not the dominant infiltrating cell type in muscle and skin biopsies in DM and few cytotoxic cells were present in infiltrates which may suggest divergent pathologies between DM and other CTDs.

### 4.9. Patient Stratification: Predictive Powers of CD8^+^ T Cells

The inherent heterogeneity of clinical phenotypes between and within CTDs poses great challenges in interpretation of phenotyping and clinical data from variable patient cohorts. Machine learning approaches are becoming an increasingly useful tool to stratify heterogeneous groups of patients to predict diagnosis and disease prognosis with the aim of supporting a shift towards personalised medicine (reviewed in [173]). Successful application of such approaches has shown that frequencies of CD8^+^ T cell phenotypes in the blood can predict disease trajectories in SLE, JSLE, and SS. In SLE a group of patients with expanded CD8^+^ memory populations was associated with lower flare-free survival time [64]. Similarly, JSLE patients with elevated CD8^+^ T cells and CD8^+^ EM cells had more active disease over time and more renal involvement [93]. Moreover, a recent analysis using K means clustering of mixed patients (pSS, SLE, and SLE/SS) revealed a group with divergent CD8^+^ T cell populations: decreased CD8^+^ CD127^+^CD25^−^, CD8^+^ CM, CD8^+^ naïve, and increased CD8^+^ TEMRA, CD8^+^CD25^−^CD127^−^, CD8^+^ T EM and total CD8^+^ T cells. Patients in this group had increased disease activity and distinct serological markers at baseline and damage scores over time. Thus, through machine learning stratification approaches frequencies of CD8^+^ T cell populations have emerged as an important predictor of disease progression in CTDs [73].

Although this review focused on differences between disease phenotypes and matched controls, it is worth mentioning that CD8^+^ T cell signatures have also been reported in several papers exploring within disease patient stratification. In JSLE alterations in CD8^+^ T cell phenotype and transcriptome have been associated with cardiovascular disease (CVD) risk [174]. Furthermore, in myositis major differences in the CD8^+^ T cell transcriptome were identified between patients with PM and DM, suggesting their involvement in the disease pathogenesis [130].

Research aiming at molecular stratification of patients with CTDs for selection of therapies in clinical trials has the potential to identify various molecular signatures and establish the role of CD8^+^ T cells in personalised medicine approaches. Significant progress has been achieved in the molecular characterisation of patients with SLE, SS, and SSc based on Type 1 and 2 IFN molecular signatures, peripheral blood immune cell phenotypes, single nucleotide polymorphisms, and methylation status [175,176,177].

### 4.10. Targeting CD8^+^ T Cells in CTDs

The pathogenesis of CTDs is diverse and in the absence of complete understanding of the underlying pathology there is currently no cure. However, growing evidence of CD8^+^ T cell dysregulation in CTDs makes them an attractive target for future therapies. CD8^+^ T cell targeted therapies for treatment of autoimmune disease must inhibit autoreactive cytotoxic T cells, while at the same time, maintaining their cancer immunosurveillance and infection protection functions.

In addition to antigen presentation via the MHC-I complex, CD8^+^ T cell activation also requires a costimulatory signal via binding between CD80/CD86 on the APC to CD28 on the T cell surface. Abatacept is a fusion protein composed of the extracellular domain of CTLA-4 and Fc portion of immunoglobulin (Ig) and it can block T cell activation by interfering with the costimulatory signals by binding to CD80/CD86 [178]. Abatacept has shown some promise in clinical trials of CTDs. In a proof-of-concept study, Meiners et al. showed that abatacept was effective, safe, and well tolerated in patients with pSS and resulted in reductions in ESSDAI and EULAR Sjögren’s syndrome Patient Reported Index (ESSPRI) [179]. Another clinical trial involving patients with sSS and RA also showed abatacept efficacy with significant decreases in disease activity in SS and RA patients [180]. However, two subsequent placebo-controlled trials found abatacept to perform no better than placebo in active pSS [181,182]. 

In SSc costimulatory blockade with abatacept has been shown to be safe and effective in reducing joint involvement and related disability [183]. In a phase 2 placebo-controlled trial abatacept treatment also improved general health in patients with early dcSSc and showed a trend for skin improvement in the abatacept treated group [184]. Abatacept therapy has also shown promise myositis; in PM and DM patients treatment resulted in lower disease activity in 8 out of 19 patients [185] and CD4/CD8 T cell ratio in peripheral blood, driven by alterations in CD8^+^ T cells, correlated positively with muscle endurance improvement [186]. Although all clinical trials of abatacept in SLE thus far have failed their clinical end points [187], abatacept has been shown to be effective in preventing articular flares and improving quality of life and fatigue in patients with non-life threatening disease manifestations [188] and improved serological activity and achieved greater reduction in protein: creatinine ratio in patients with LN compared to placebo [189], suggesting a possible role for costimulatory blockade in a subset of SLE patients.

It should be noted that only one of these trials [186] included CD8^+^ T cell population detection, thus it is not possible to determine the effect of abatacept therapy on CD8^+^ T cells. Further studies are necessary to determine disease-specific effects of abatacept on CD8^+^ T cells in CTDs.

Other potential therapies aimed at modulating CD8^+^ T cell effector function include IFN-γ blockade and granzyme B blockade. IFN-γ blockade with AMG811, a human monoclonal antibody against IFN-γ, initially showed some promise in SLE where treatment of stable SLE patients, led to dose dependent modulation of genes associated with IFN-γ signalling and reduction in serum levels of CXCL10, a key chemokine associated with SLE disease activity [190]. However, no clinical efficacy was demonstrated in a subsequent study in patients with and without LN [191]. 

Granzyme inhibitors such as serine protein inhibitor A3N (serpina3n) and VTI-1002 have demonstrated efficacy in animal models of multiple sclerosis (MS) and autoimmune blistering disease. Serpina3n administration in an MS mouse model reduced disease severity and reduced axonal and neuronal injury [192] while granzyme B pharmacological inhibition with VTI-1002 ameliorated disease in 3 murine models of autoimmune blistering disease [193]. As yet, there is no evidence of granzyme B blockade efficacy in human studies or in mouse models of CTD.

Therapies aimed at modulating CD8^+^ T cell function in CTDs, must strike a balance by dampening the unwanted systemic and local effector response without compromising normal CD8^+^ T cell function in infection response and cancer surveillance. There is a strong bidirectional link between cancer and autoimmunity, although the precise details of this relationship remain to be elucidated. Increased risk of some malignancies has been observed in SLE, SS, SSc, and DM [194,195,196,197]. It is unclear whether the increased incidence of cancers is due to pathological autoimmune processes, long-term immunosuppressive therapy, or other factors. Moreover, in SSc there is some evidence foreign tumour antigens may actually trigger the autoimmune response [198]. 

Immune checkpoint inhibitor therapy has shown promise in several malignancies [199]. Immune checkpoints are accessory molecules that enhance or inhibit the T cell response. Tumour cells often upregulate checkpoint proteins as a protective mechanism to shield them from immune attack. Immune checkpoint inhibitors are monoclonal antibodies that interfere with the checkpoint protein: receptor interactions, leading to enhanced CD8^+^ T cell activation and cytotoxic response against malignant cells [200]. The finding that tumour antigens may trigger SSc suggests a possible role for checkpoint inhibitor therapy in a subset of SSc patients, however, the role or tumour antigens in perpetuating the disease is unclear as the cancer was eradicated in most patients by the time SSc developed [198], thus checkpoint inhibition may be of limited efficacy in established SSc. 

In theory, checkpoint inhibition therapy may be an attractive option in a subset of SLE patients with diminished CD8^+^ T cell function [102] as boosting defective CD8^+^ T cell responses could enhance infection protection and cancer immunosurveillance in these patients. However, it may also lead to exacerbation of the disease and have unwarranted side effects. Although only two SLE patients were included in this study, administration of ipilimumab (antibody against CTLA-4) to treat advanced melanoma in patients with pre-existing autoimmune diseases was associated with higher rates and severity of adverse events than general cancer populations and 27% of patients experienced exacerbation of their autoimmune condition [201].

### 4.11. Regulatory CD8^+^ T Cell Therapies

Current treatments for CTDs rely on broad immunosuppression with corticosteroids and disease modifying antirheumatic drugs. The unwarranted side effects caused by lack of specificity of these treatments could be alleviated with antigen-specific approaches. Regulatory CD8^+^ T cells which suppress CD4^+^ T cells or APCs hold much promise in this regard as they could be engineered to suppress the pathogenic immune response to specific autoantigens [202]. CD8^+^ Tregs can be induced in vitro using a variety of cell culture conditions to develop into the CD8^+^ Treg phenotype and CD8^+^ T cell specificities can be manipulated using TCR and chimeric antigen receptor (CAR) engineering [203]. The desired CD8^+^ T cells could then be transferred back into the patient either systemically or injected into organs affected by autoimmune disease. Challenges of this type of therapy include limited lifespan of such cells, the necessity to adapt the therapy to individual hosts (as transfer of allogeneic cells would not be acceptable due to potential side effects), and the fact that specific antigens in many CTDs have not yet been identified. It is also not known if Treg suppression would provide permanent or transient inhibition of immune response [202]. Research focusing on methods of growing large numbers of CD8^+^ Tregs and exploring their potential use in localised inflammation is ongoing. Moreover, the first in human phase I CD8^+^ Treg therapy in kidney transplant patients is underway and results from this trial may pave the way for future CD8^+^ Treg therapy in CTDs [203]. As some CTDs are associated with increased risks of malignancies, (notably SS patients have a 4.3-fold increased risk of Non-Hodgkin’s Lymphoma compared to the general population) [204], any future Treg therapies must balance the risk of Treg inhibition of anti-tumour responses with the clinical benefit of controlling autoimmune disease. 

## 5. Conclusions

There is growing evidence of aberrations in CD8^+^ T cell phenotype and function in CTDs. However, it is as yet unclear to what extent CD8^+^ T cells contribute to the pathology and progression of these diseases or if aberrations in CD8^+^ T cell phenotype and function are a consequence of systemic immune dysfunction or immunosuppression therapy. Human studies examining the CD8^+^ T cell immunophenotype in peripheral blood in CTDs are hampered by the inherent heterogeneity of patient cohorts and may benefit from patient stratification approaches. Cytotoxic CD8^+^ T cell infiltration of affected organs is a common feature of CTDs, and targeted treatments aimed at inhibiting CD8^+^ T cell function at local sites of inflammation may be of therapeutic benefit. Further research is necessary to help us better understand the role CD8^+^ T cells play in CTDs and how to manipulate these cells for clinical benefit.

## Figures and Tables

**Figure 1 ijms-23-11431-f001:**
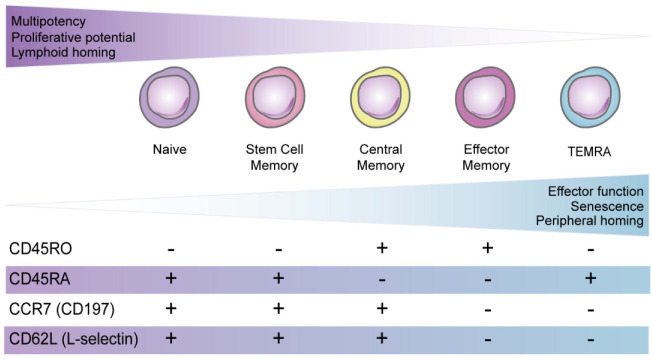
Differentiation markers characterizing various CD8^+^ T cell populations. Changes in surface marker expression of CD45RO, CD45RA, CCR7 and CD62L define stages of CD8^+^ T cell differentiation from naïve to terminally differentiated effector memory (TEMRA) CD8^+^ T cells. Gradient triangles indicate preferential functions of CD8^+^ T cell subpopulations.

**Figure 2 ijms-23-11431-f002:**
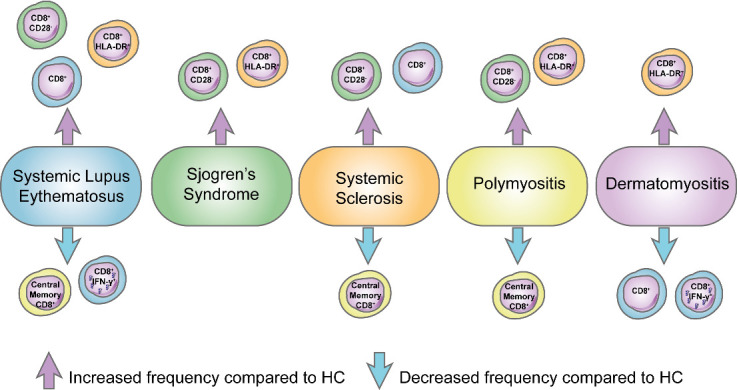
Shared aberrations in peripheral blood CD8^+^ T cell population frequencies across adult CTDs compared to HC. In DM, frequencies of total CD8^+^ T cells as well as CD8^+^ T cells expressing intracellular IFN-γ were only diminished in patients with active disease.

**Table 1 ijms-23-11431-t001:** Overview of CD8^+^ T cell effector populations and their known functions.

Subset	Effector Molecules	Transcription Factors	Function	Cytotoxicity	Ref
Tc1	IFN-γ, TNF-α, perforin, granzyme B	Tbet, EOMES, STAT4	Immunity against intracellular pathogens and tumours	Yes	[7,8]
Tc2	IL-4, IL-5, IL-13, granzyme B	GATA-3, STAT6	Maintenance of allergy responses	Yes	[8,9,10]
Tc9	IL-9	IRF-4, STAT6	Maintenance of allergy responses	No	[11,12]
Tc17	IL-17, IL-21, IL-22	RORyT, IRF-4, STAT3	Propagation of autoimmunity, immunity against fungal pathogens, anti-tumour response	No	[13,14,15]

Abbreviations used: EOMES = eomesodermin, GATA-3 = GATA-Binding Factor 3, IFN = interferon, IL = interleukin, IRF = interferon regulatory factor, ROR = RAR-related orphan receptor, STAT = signal transducer and activator of transcription, Tbet = T-Box expressed in T cells, Tc = cytotoxic T cell, TNF = tumour necrosis factor.

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
