# Peer review of "CD8+ T Cell Phenotype and Function in Childhood and Adult-Onset Connective Tissue Disease"

_ijms, 2022, doi:10.3390/ijms231911431_

Round 1

Reviewer 1 Report

Radziszewska et al provide an overview of the literature surrounding CD8+ T cells and connective tissue disease (CTD).  The manuscript is extensive in its review and discussion of current evidence supporting a role for CD8+ T cells in numerous CTDs and presents interesting literature describing CD8+ T cell functional abnormalities associated with CTD. This review would be of interest to the larger field. However, I do have several comments:

11. In Figure 1, the diagram indicates that Tcm lack CD62L expression. This is incorrect (Sallusto et al 1999).

22. Tissue-resident memory CD8+ T cells (Trm) have recently been described as an important subset contributing to health and disease (including autoimmune disease). While the manuscript discusses organ-specific CD8+ T cells, this subset is not included in the introduction and only acknowledged one time in the text (line 419). The exclusion of Trm in a discussion of CD8+ T cells and particularly subsets of CD8+ T cells makes the manuscript seem incomplete.  

33. The organization of the manuscript is a bit confusing.

·         Despite the section title, Section 1.1 is really CD8+ T cells in health then the subsequent sections 1.2-1.5 are sections relating to autoimmunity. It might be better organized as 2 sections, with the specific diseases being subsections of the latter.

·         The Aims and Methodology could be the second section instead of a sub-section under Introduction.

·         The results section is dense – in particular the sections describing the phenotypic alterations. It might help the reader if it was divided in subsection based on the specific disease (as in the introduction).

·         Some of the discussion in the Result’s and Discussion’s section seem to overlap and could be streamlined to help the reader.

44. On line 421-422, the authors state the CD8+ T cell exhaustion does not occur in the kidney as kidney-specific T cells express low levels of exhaustion markers compared to peripheral blood T cells. This is an overstatement of the evidence presented and should be reworded. Functional status of a T cell can not necessarily be concluded from phenotypic markers, especially from one study examining one disease state.

Minor Comments:

11. It might be helpful to include references in Table 1 (as you do in the other Tables in your manuscript).

22. Line 97-98 reads that CD8+ T cells are producing autoantibodies. This should be clarified.

33. The text size between your Tables is very variable.

44. The subsection title on line 332-333 is incorporated into the footnotes of your Table 4.

55. Paragraph on lines 447-451 seems incomplete and ends mid-sentence.

Reviewer 2 Report

In this review, the authors have attempted to provide an updated, comprehensive account of the types of cytotoxic T cells (Tc) and their function in various connective tissue diseases (CTDs).

I disagree with the term “phenotypes” as this alludes to detailed phenotypic analyses of Tc cells in peripheral blood or in situ, which are not included in this work. Figure 1 is misdescribed, i.e., it does not show Tc cell heterogeneity but rather certain Tc differentiation markers.

The authors should strive to present more detailed information on markers that distinguish between effector and regulatory Tc cells and the cytokines they produce.

Table 1 is mislabeled because it includes CD8+ Tregs. It should also include the relevant references. In addition, information on the expression of Foxp3 by CD8+ Tregs should be added (see, for example, Liu et al, J Mol Cell Biol. 2014, 6(1):81-92).

Section 1.3., Table 3, and elsewhere in the text lack pertinent information about CD8+CD28- Tregs in SS (see, for example, Psianou et al, Autoimmun Rev. 2021, 17(10):1053-1064).

In section 3.10., the authors assume that the role of CD8+ Tregs in CTDs is beneficial, and could potentially be used for therapy, and they refer to a clinical trial using CD8+ Tregs in kidney transplantation (in fact, to a review paper mentioning this [185]), the results of which will “pave the way” for similar trials/studies in CTD patients. This is an unfounded assumption, because in certain CTDs, especially SLE and SS, there is increased incidence in the development of hematologic malignancies, and, actually, SS has the highest risk of lymphoma development among autoimmune diseases (see, for example, Kapsogeorgou et al, J Autoimmun. 2019, 104:102316). Thus, given the deleterious role of Tregs in neoplasms, there is no basis for such an assumption. 

Round 2

Reviewer 2 Report

It seems that Fig.1 is placed 2ce in the revised manuscript. If this is so, remove 1.